# Efficacy of metamizole versus ibuprofen and a short educational intervention versus standard care in acute and subacute low back pain: a study protocol of a randomised, multicentre, factorial trial (EMISI trial)

Maria M Wertli  ,[1,2] Julian S Flury,[1,2] Sven Streit  ,[3] Andreas Limacher,[4] Vanessa Schuler,[1,2] Asha-Naima Ferrante,[1,2,5] Caroline Rimensberger,[1,2] Manuel Haschke[2,6]

For numbered affiliations see end of article.

**Correspondence to**
Professor Maria M Wertli; maria.wertli@insel.ch

## ABSTRACT

**Introduction** Low back pain (LBP) is among the top three most common diseases worldwide, resulting in a life with pain-related disability. To date, no study has assessed the efficacy of metamizole (dipyrone), a non-opioid analgesic and antipyretic prodrug compared with the conventional non-steroidal anti-inflammatory drug ibuprofen, in patients with an acute LBP episode. Further, it is unclear, whether a short educational intervention is superior to usual care alone.

**Objectives** The objective of this study is to assess first, whether metamizole is non-inferior to ibuprofen in a new episode of acute or subacute LBP. Second, we aim to assess whether a short educational intervention including evidence-based patient information on the nature of LBP is superior to usual care alone.

**Methods and analysis** An investigator-initiated multicentre, randomised, double blind trial using a factorial design will be performed. A total of 120 participants with a new episode of LBP will be recruited from GP practices, outpatient clinics and from emergency departments, and randomised into four different treatment groups: ibuprofen alone, ibuprofen and short intervention, metamizole alone, metamizole and short intervention. The primary endpoint for the medical treatment will be change in pain assessed on an 11-point Numeric Rating Scale after 14 days. The primary outcome for the short intervention will be change in the Core Outcome Measures Index assessed after 42 days.

**Ethics, dissemination and funding** This study has been approved by the responsible Ethics Board (Ethikkommission Bern/2018-01986) and the Swiss Agency for Therapeutic Products (Swissmedic/2019DR4002). Results will be published in open access policy peer-reviewed journals. The study is funded by the Swiss National Science Foundation (grant number 32 003B-179346).

**Trial registration number** NCT04111315

## Strengths and limitations of this study

► This is the first randomised double-blind study assessing the analgesic effect of metamizole compared with the conventional non-steroidal anti-inflammatory drug ibuprofen in patients with acute or subacute low back pain (LBP) presenting to the GP or emergency department.

► In addition, this study will assess whether a short educational intervention is more effective in patients with LBP compared with usual care.

► Due to the high proportion of patients who will not require extended pain treatment during an acute LBP episode, the trial design mirrors current clinical practice.

► The primary endpoint pain intensity will be assessed after 14 days. The primary outcome for the short intervention will be assessed after 42 days.

► While patients, physicians and researchers are blinded to the pharmacological treatment, blinding is not feasible for the short educational intervention.

## INTRODUCTION

Low back pain (LBP) is among the top three most common diseases worldwide, resulting in a life with pain-related disability.[1] In patients with persistent LBP over 3 months, the risk for chronic pain increases sharply[2] and effective interventions should aim to prevent pain persistence without overtreatment.[3] Therefore, treatment guidelines recommend the use of pain medication and to keep patients physically active during an acute LBP episode.[4 5] Despite the frequency of LBP, only few high-quality studies assessed the efficacy of pain medication. Systematic reviews of randomised clinical trials found

opioids to be no more effective than non-opioid medication in LBP[6 7] but opioids were associated with potentially severe side effects.[4 7 8] Therefore, non-opioid pain medications that are safe and effective are urgently needed to control pain.

Current treatment guidelines recommend non-steroidal anti-inflammatory drugs (NSAIDs) for acute pain control.[4 9] However, adverse events (AEs) including kidney injury and gastrointestinal bleeding limit their use in many patients. Metamizole (dipyrone) is a non-opioid analgesic and antipyretic with a favourable gastrointestinal and renal safety profile.[10] For example, metamizole use was associated with a lower risk for gastrointestinal bleeding compared with NSAIDs and for hepatotoxicity compared with paracetamol.[11] Although metamizole has been withdrawn from the market or never approved in many countries including the USA, Canada and UK,[12] it is frequently used in many other countries such as Germany,[13] Spain,[12] Switzerland,[14] some Eastern European and South American countries.[12 15] Despite its increasing use in many European countries,[14 16] the role of metamizole for the treatment of LBP is unclear and has so far not been systematically studied. This may be due to an ongoing controversy regarding the risk of metamizole-associated agranulocytosis.[12 15] Due to the rare occurrence, incidence rates have been based on case–control studies and pharmacovigilance data. The incidence of the metamizole-induced agranulocytosis was estimated between 0.5 and 3.5 cases per 1 million metamizole users over 1 week per year.[17–28] Even though the overall risk of agranulocytosis is higher compared with other analgesics, it remains a rare event occurring only in a small proportion of susceptible patients[17–25 27–29] and the overall safety profile of metamizole is still favourable compared with NSAIDs or opioids.[30]

Many patients have reservations against regular intake of pain medication and, contrary to guideline recommendations, limit physical activity to keep pain manageable without medication.[31] Patient education encouraging physical activity may help to improve pain control and function. For example, a Cochrane review found that 2.5-hour educational sessions were more effective than no intervention on the return-to-work rate in patients with acute/subacute LBP.[32] Such time-intensive interventions may not be necessary in all patients and are not feasible in a busy primary care practice. Short educational interventions would be more practical, however, to date, it is unclear whether they also have a positive effect on outcome in patients with a new LBP episode.

## Objectives
The first objective of this study is to assess whether metamizole is non-inferior to ibuprofen in patients with a new episode of acute or subacute LBP. The second objective is to assess whether a short educational intervention, including evidence-based patient information on the nature of LBP, is superior to usual care alone. Thus, this randomised trial will provide new knowledge regarding the efficacy of non-opioid pain medications in LBP. In addition, if the short educational intervention proves to be beneficial compared with usual care, the trial will help to improve care for patients with a new LBP episode.

## METHODS AND ANALYSIS
The methods reporting of this randomised trial will follow the recommendations of the Standard Protocol Items: Recommendations for Interventional Trials (SPIRIT) statement.[33]

### Design: multicentre, randomised, double-blind trial using a factorial design
The Efficacy of Metamizole versus Ibuprofen and a Short Educational Intervention versus Standard Care in Acute and Subacute Low Back Pain (EMISI) study is a multi-centre, randomised, double-blind trial using a factorial design to investigate whether metamizole is non-inferior to ibuprofen in patients with a new-onset LBP episode. Patients, physicians and researchers will be blinded to the allocation of the study medication. A non-inferiority design was chosen as metamizole seems to be equally effective as ibuprofen in clinical practice. Furthermore, the study will assess whether a short educational intervention including evidence-based patient information is superior to usual care. Blinding patients about the short intervention is not possible but blinding is upheld in GPs and researchers (double blind). The primary evaluation of outcomes (see the Outcome measure section for details) will be performed at day 14 (comparison of metamizole to ibuprofen) and at day 42 (comparison of educational short intervention to usual care).

### Eligibility criteria
Subjects fulfilling all of the following inclusion criteria are eligible for the study:
► Age 18 years or older.
► Seeking care for new onset of non-specific or specific LBP (pain duration of less than 12 weeks LBP prior to the baseline visit).
► The GP plans to prescribe a non-opioid pain medication for pain control.
► Informed consent documented by signature.
  Exclusion criteria are:
► Chronic use of opioids.
► Presence of red flags (serious neurological deficit requiring surgery, infection, vertebral fracture).
► Known intolerance to the study medication (ie, previous acute allergic reaction to NSAIDs or metamizole).
► Active malignancy and/or history of a (previous) haematological disorder (history of agranulocytosis).
► History of anaemia (haemoglobin of <10.0 g/L), neutropaenia (leucocyte count of <3.0 x $10^9$/L), thrombocytopaenia (<100 G/L).
► Known contraindications against the study medications: previous gastrointestinal ulcer/bleeding,

inflammatory bowel disease, heart failure (New York Heart Association (NYHA) Functional Classification III–IV), liver failure (liver cirrhosis, ascites), renal insufficiency (estimated glomerular filtration rate (eGFR) <60 mL/min/1.73 m$^2$) or previous acute kidney injury (AKI stage 2 according to the Kidney Disease: Improving Global Outcomes (KDIGO) definition).

► Immune deficiency or under immunosuppressant treatment.
► Known or suspected non-compliance, drug or alcohol abuse.
► Participation in another study within 30 days preceding randomisation and during the present study or previous enrolment into the current study.
► Enrolment of the investigator, his/her family members, employees and other dependent persons.
► Pregnancy: In women of childbearing age, a negative pregnancy test (urine or blood test as available in the primary care practice) before inclusion is required. Women who are not willing to use safe contraception (condom or birth control pill) during the trial, intention to become pregnant during the trial, pregnancy or breast feeding.

### Study setting

The EMISI trial will be coordinated by the central study team at the Division of General Internal Medicine, University Hospital Bern, Switzerland. Patients will be recruited from outpatient clinics, from GP practices, and from emergency departments. The overall study duration of 42 days (6 weeks) was chosen because treatment guidelines recommend further assessment of patients for multidisciplinary treatment if LBP persists for more than 6 weeks despite pain medication and the advice to stay active. This is an investigator-initiated trial. The funder will have no role in the study design, the trial oversight, data collection, analysis of the study and publication of the results.

### Estimated sample size and power

Both interventions are considered for determining the sample size for this factorial trial.[34] We expect the two primary hypotheses to be independent of each other and do not expect an interaction between both interventions. Therefore, we powered each hypothesis individually, but accounted for multiple testing by adjusting the type-I error rate by setting the alpha level to 0.025 in each test to keep the overall type-I error rate at 0.05. Thus, we aimed to include 120 patients to account for a drop-out of 10% which will allow to proof non-inferiority of the metamizole treatment at a one-sided alpha-level of 2.5% with a power of 90%.

The sample size calculation for the comparison of metamizole versus ibuprofen was based on a difference in the change of pain score in the Numeric Rating Scale (NRS; range 0–10, higher score indicates more pain). We considered a change of ≥2 points between the two

groups as clinically relevant and a change of ≤1 point as negligible.[35 36] Therefore, we set the non-inferiority margin to −1 score point. Based on a two-sample means test and an SD of 1.6,[35] we will need 108 patients (54 per group) to proof non-inferiority of the metamizole treatment at a one-sided alpha-level of 2.5% with a power of 90%.

The sample size for the primary outcome comparison short intervention versus usual care (superiority) is based on a change in Core Outcome Measures Index (COMI; range 0–10, higher score indicates higher level of complaint, see primary outcomes) from baseline to week 6. We hypothesise that there is a difference between the two intervention groups regarding COMI. Mannion et al[37] reported a minimal clinically important difference (MCID) for improvement in the COMI of 2.2 points and SD for changes in COMI ranging from 1.7 to 2.5 points. We see a difference of 2.2 points between the two groups as clinically relevant and assume a SD of 2.2 points. Based on a two-sample means test, we will need 54 patients (27 per group) to detect a difference in the change of the COMI at a two-sided alpha-level of 2.5% with a power of 90%.

### Primary outcome

Table 1 provides an overview of measurements according to the SPIRIT. The primary outcome for the comparison of metamizole versus ibuprofen is the change in pain on the NRS (range 0–10 points) from baseline to follow-up at day 14 (non-inferiority). Pain intensity will be recorded as average pain intensity for the past 24 hours on an NRS scale from 0 to 10, scored from 0 (no pain) to 10 (worst possible pain). The rational for assessing the effectiveness outcome at 14 days was based on the double-blind RCT comparing paracetamol to placebo.[38]

The primary outcome for the comparison of the short intervention versus usual care is the change in the COMI sum-score (range 0–10) from baseline to 42 days follow-up (superiority). The COMI is a short, multidimensional, validated outcome instrument with excellent psychometric properties recommended for the monitoring of the outcome in patients undergoing treatment for LBP.[37] The COMI sum score is calculated as described in the validation paper.[39 40] The pain scales (item 2a and 2b) are scored 0–10, while the category scales (items 3–7) are scored as 0, 2.5, 5.0, 7.5, 10.0. The sum score (0–10) for the whole COMI scale is calculated based on the following five domains:

► Pain domain: higher of the two pain scores (higher of 2a and 2b) is taken as the score.
► Disability domain: the average of the two disability items (6 and 7).
► Back related function (item 3).
► Symptom specific well-being (item 4).
► General quality of life (item 5).

The five domain scores are then averaged to give a COMI score from 0 to 10.

**Table 1** Summary of study procedures based on Standard Protocol Items: Recommendations for Interventional Trials schedule

| | Enrolment | Allocation | Short intervention | Phone call | Clinical visit | Phone call | Clinical visit or phone call |
|---|---|---|---|---|---|---|---|
| Time point (days) | −5 to 0 | 0 | 2 | 7 | 14 | 28 | 42 |
| Clinical visit | 1 | 1 | | | 2 | | **3 or** ✆[4] |
| Phone call | | | ✆[2] | ✆[3] | | ✆[3] | |
| Patient information and informed consent | x | | | | | | |
| Inclusion/exclusion | x | | | | | | |
| Demographics | x | | | | | | |
| Medical history | x | | | | | | |
| Physical examination | x | | | | | | |
| Vital signs | x | | | | x | | (x) |
| Blood analysis | x | | | | x | | (x) |
| Pregnancy test | x | | | | | | |
| Randomisation | | x | | | | | |
| Hand out study medication | | x | | | | | |
| Daily intake of study medication | | x | | | | | |
| Group 1+3: study medication only | | x | | x | x | x | x |
| Group 2+4: study medication +short intervention | | x | x | x | x | x | x |
| Pain diary daily including:<br>► Pain<br>► Additional treatments<br>► AE<br>► Daily physical activity | | x | x | x | x | x | x |
| Adherence reminder[3] | | | | x | | x | |
| Screening for[9]:<br>► AE and SAE<br>► Interrupting or discontinuing treatment | | | | x | x | x | x |
| Collection of pill bottles for pill count | | | | | x | | x |
| Self-reported measures collected during visits | | | | | | | |
| Pain questionnaire (NRS) | | x | | | x | | x |
| COMI | | x | | | x | | x |
| STarT Back tool | | x | | | x | | x |
| Quality of life (EQ5-D-5L) | | x | | | x | | x |
| Fear avoidance (FABQ) | | x | | | | | |
| Self-efficacy (FESS) | | x | | | | | |
| Blinding control | | | | | | | x |
| Voluntary in-person interview on patient perspective | | | | | | | x |

AE, adverse event; COMI, Core Outcome Measures Index; EQ5-D-5L, European Quality of Life-5 Dimensions questionnaire; FABQ, Fear Avoidance Questionnaire; FESS, FESS, self-efficacy (Schmerzspezifische Selbstwirksamkeit); NRS, Numeric Rating Scale; SAE, serious adverse event.

Two additional questions cover the satisfaction with treatment outcome. An MCID found to be relevant for patients was a reduction in COMI score of ≥2.2 points.[37]

## Secondary outcome

All outcome measures used in this study were recommended by the NIH Task force and other expert panels to study back pain.[41 42] Table 1 provides an overview of measurements according to the SPIRIT.[33]

Secondary outcome measures include:

► Change in pain on the NRS (range 0–10 points) from baseline to follow-up at day 42.
► Change in the COMI sum-score (range 0–10) from baseline to 14 days follow-up.
► The proportion of patients with an MCID in the NRS (≥2 points) and COMI (≥2.2 points) at days 14 and 42.
► Time to recovery (recovery defined as the first day with an NRS of <2 points sustained for two consecutive days).
► Change in overall pain intensity from baseline to follow-up at day 14 and day 42. Overall pain intensity: average of ratings for average pain, pain at rest, and pain during activity during the last 24 hours.
► Area under effect curve (day 0–14) NRS.
► AEs.
► Time to stopping treatment.
► Global effectiveness of the treatment (COMI question 'how much did the operation/treatment help your back problem?' 5-point Likert scale from 'helped a lot' to 'made things worse').
► Daily physical activity (daily step count).
► Use of pain medication and additional rescue medication.
► Additional treatment (physical therapy, injections).
► Return to work (yes/no).
► Treatment satisfaction (COMI).

► Quality of life as assessed by the 5-level version of the European Quality of Life-5 Dimensions questionnaire (EQ-5D).
► Psychological factors (STarT-Back Tool).[43]

## Exploratory variables

Response to pain medications may be influenced by individual pharmacokinetic and pharmacodynamics factors. Therefore, we will collect blood samples during visit 2 (day 14) to assess the following factors that may influence the individual response to the pain medications. First, we will measure plasma concentrations of ibuprofen and the four main metamizole metabolites (4-MAA, 4-AA, 4-FA, 4-AAA). Pharmacodynamic effects of metamizole correlate primarily with concentrations of the 4-MAA metabolite,[44] thus, direct quantification of 4-MAA concentrations will add important information for the interpretation of the observed effects.[45]

Second, as individual response to ibuprofen may be affected by genetic polymorphisms of cytochrome P450 (CYP) isoforms 2C8 and 2C9, the study population will be characterised in terms of CYP 2C8/9 genotypes (CYP2C8 *1 and *3 alleles, CYP2C9 *1, *2, and *3 alleles). In subjects with alleles conferring low CYP2C8 and/or CYP2C9 activity, reduced ibuprofen clearance of both enantiomers has been described, resulting in several-fold increased exposure and prolonged half-life compared with subjects with wild-type alleles.[46 47]

## Study procedures

Study procedures are summarised in the study flow chart (figure 1). Each participating site will receive a supply of sequentially coded, blinded and sealed intervention packs. All patients presenting with LBP will be screened for potential inclusion into the study. After obtaining informed consent by the physician, patient information will be entered in a web-based clinical data management

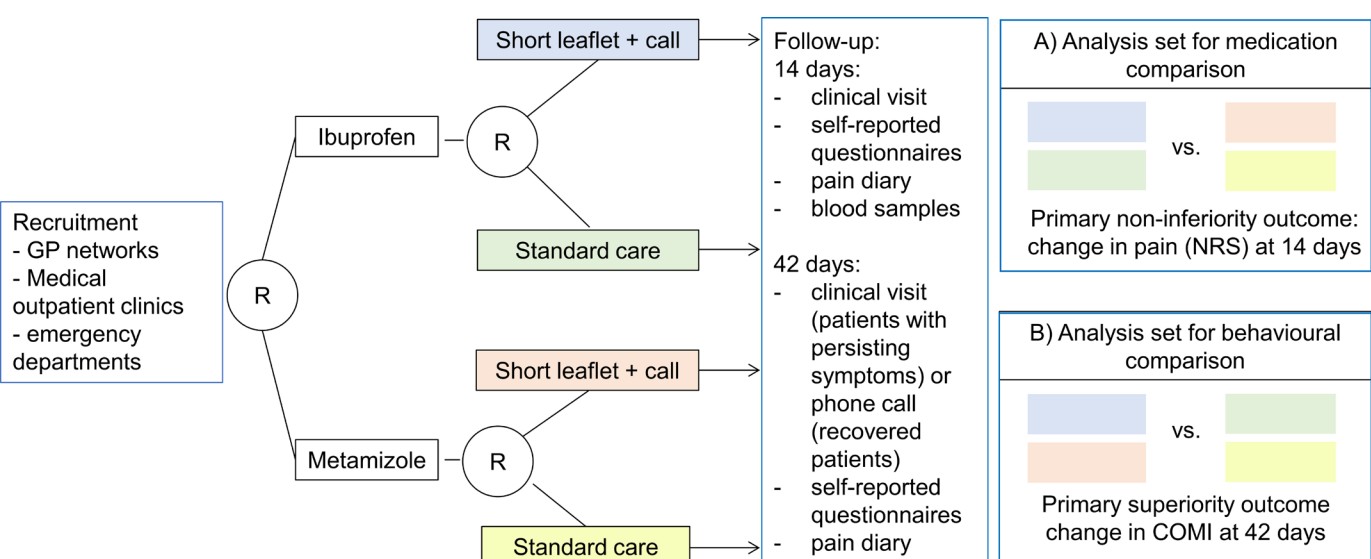

**Figure 1** Study flow chart. GP, general practitioner; COMI, Core Outcome Measures Index; NRS, Numeric Rating Scale.

system (CDMS). After patient registration in the CDMS and confirmation of all inclusion and exclusion criteria, patients are randomised using sequentially coded and blinded intervention packs, thus ensuring concealment of allocation. The number of the intervention pack will be recorded in the CDMS.

Patients will complete a set of self-reported questionnaires (see table 1 for details) on demographics (age, sex, work and civic status), previous history, current medications, pain related information (NRS, pain drawing), disability (COMI, STarT Back tool), quality of life (EQ5-D-5L), the Fear Avoidance Questionnaire,[48] self-efficacy (Schmerzspezifische Selbstwirksamkeit[49]). Physicians will complete a questionnaire on clinical findings, the clinical diagnosis, previous and newly prescribed treatments.

On completion of the baseline evaluation, patients may open the intervention pack and start treatment. The intervention pack will contain a pill bottle and an envelope with the assignment of the short versus no intervention and a pain diary.

Between the first and the second visit, patients will record their pain before intake of study drug, the number of study drugs and additional pain medications, and their daily physical activity (quantified by daily step counts using a commercially available wearable step tracking device (Garmin vivofit 4).

To mitigate the risk for AEs, we will collect blood samples after the randomisation in all patients. In case of a newly detected renal insufficiency (defined as eGFR $<60\,\mathrm{mL/min/1.73\ m^2}$), leucopaenia (leucocyte count $<3.0\,\mathrm{G/L}$), thrombocytopaenia (defined as $<100\,\mathrm{G/L}$) and/or anaemia ((haemoglobin $<100\,\mathrm{g/L}$), we will stop the study medication.

During the second visit (day 14), patients will return the pill bottles and the pain diary to the GPs office where they are collected for pill count. The patient will receive a new drug pack and pain diary. A blood sample will be collected (creatinine, blood count, drug concentrations and genotyping), and self-reported questionnaires and a clinical assessment will be completed. The GP will record the clinical findings and additional pharmacological and non-pharmacological treatments.

To improve adherence, study staff will call patients between the first and the second visit and between the second and the final visit and ask for AEs and additional pain treatment. As we expect a high proportion of patients to have recovered by day 28 (second phone call), researchers will then determine whether a clinical visit on day 42 is necessary. In case patients have recovered by day 28, patients do not need to attend the clinical visit. Patients will be asked whether they would be willing to participate in an in-person interview at the end of the study. The aim of the interview is to assess factors that were relevant for patients during their LBP treatment. The interview will be recorded (video and audio) for qualitative analysis using standard grounded theory method according to Philipp Mayring.[50]

A final assessment will be done on day 42 during a clinical visit in patients who had persisting symptoms on day 28. Patients will return the pain diary and the pill bottles with the remaining pills to the GPs office where they are collected for pill count. Data collection includes self-reported validated questionnaires. The GP will record the clinical findings and additional pharmacological and non-pharmacological treatments. A final blood sample (serum creatinine and blood count) will only be collected if patients are still taking study medication. Patients who do not need to attend the clinical visit will receive a final identical set of questionnaires by mail and will be asked for AE/SAE and additional treatments during a phone call. Patients will return the questionnaires together with the pain diary and the pill bottles to the central study team. To assess the success of blinding, patients will be asked at the end of the study what treatment group they believe they had been assigned to (metamizole or ibuprofen).

### Pharmacological intervention

During the first 4 days, we will compare a fixed-dose regimen of metamizole 500 mg to 500 mg ibuprofen lysine (lysine-2-(4-isobutylphenyl)-propionate, corresponds to 292.6 mg ibuprofen acid) three times two capsules per day (table 2). Ibuprofen was chosen as a comparator because it is a well studied and the most frequently prescribed NSAID in Switzerland[14] with a similar dosing interval as

**Table 2** Oral doses for study medication—fixed dose and as-needed regimen

| Group | Study medication | Fixed dose regimen (days 0–3) | As-needed regimen (after day 3): recommended reduction of the oral intake when LBP decreases |
|---|---|---|---|
| 1 | Ibuprofen only | 500 mg* 2-2-2 | 2-2-2 (pain unchanged), 1-1-1 (decreased pain; NRS <4 points), 1-0-1, 1-0-0 |
| 2 | Ibuprofen +short intervention | 500 mg* 2-2-2 | 2-2-2 (pain unchanged), 1-1-1 (decreased pain; NRS<4 points), 1-0-1, 1-0-0 |
| 3 | Metamizole only | 500 mg 2-2-2 | 2-2-2 (pain unchanged), 1-1-1 (decreased pain; NRS<4 points), 1-0-1, 1-0-0 |
| 4 | Metamizole+short intervention | 500 mg 2-2-2 | 2-2-2 (pain unchanged), 1-1-1 (decreased pain; NRS<4 points), 1-0-1, 1-0-0 |

*500 mg ibuprofenum lysinum (lysine-2-(4-isobutylphenyl)-propionate) corresponds to 292.6 mg ibuprofen acid.

metamizole. The initial dose of both medications is the currently used and recommended dose for the treatment of acute pain (common daily dose for metamizole 3000 mg (maximum daily dose 4000 mg) and 3000 mg ibuprofen lysine (maximum daily dose 4000 mg)). Ibuprofen lysine 3000 mg corresponds to 1800 mg ibuprofen acid. The ibuprofen lysine formulation was chosen to facilitate blinding of the study drug, as ibuprofen lysine 500 mg was the only ibuprofen formulation available on the Swiss market, which could be prescribed with identical number of capsules and at identical intervals as metamizole. The lysine salt of ibuprofen shows better solubility and faster absorption compared with the acid form.[44 51] However, overall extent of absorption and analgesic effect in an acute pain model did not differ from the acid form.[52]

In LBP, current clinical practice is to prescribe an initial fixed dose regimen followed by an as-needed regimen that allows patients to adjust the daily dose depending on their pain intensity. To mirror clinical practice, patients will be allowed after the first 4 days to decrease the intake of the study medication according to a dose reduction schedule (as needed phase). From day 4 until recovery, patients may reduce the daily pain medication intake once their pain has decreased to less than four points on the NRS (range 0–10 points) every 2 days. The treatment duration will thus depend on the duration of pain. Patients are considered to have recovered when their pain is below 2 (NRS 0–10) on two consecutive days after they have stopped intake of pain medication (end of treatment).

### Non-pharmacological educational short intervention

Patients assigned to the short intervention will receive together with the pain medication a short leaflet focusing on the nature of non-specific LBP and the importance of staying active. A second leaflet contains exercises to alleviate LBP. The content of the short information leaflet is based on the *The Back Book*.[53] Further, all patients assigned to the short intervention will receive a 10 min standardised telephone intervention delivered by a trained member of the study team during the first 4 treatment days to help patients understand the non-dangerous nature of LBP, and to explain the purpose of pain medication to facilitate movement. In addition, patients will be encouraged to stay active and to perform the exercises as proposed in the exercise leaflet.

### Concomitant interventions

In a double-blind randomized clinical trial (RCT) comparing paracetamol to placebo for LBP in the primary care setting, rescue medications were infrequently used.[38] Therefore, physicians will be allowed to prescribe additional rescue medications recommended by treatment guidelines at their discretion. Additional medication use is recorded in the pain diary, during the phone calls and the clinical visits. Further, information on any use of physical therapy, massage or other treatments is collected during the follow-up visits at days 14 and 42. Although GPs will be instructed to withhold physical therapy until day

14 (ie, the primary effectiveness endpoint for metamizole vs ibuprofen) if the clinical situation allows, additional care including physical therapy and relaxation strategies recommended by guidelines will be at the discretion of the treating GP. GPs will record any additional prescription of addiational care. Concomitant treatments will be treated as covariates to assess their potential impact on overall efficacy (performance bias).

### Randomisation and allocation procedures

The allocation sequence (randomisation list) will be generated by an independent data manager at the clinical trial unit (CTU) Bern who is not otherwise involved in the trial. Randomisation will be blocked and stratified according to study sites. To ensure concealment of allocation, block size will not be communicated. After patient data are registered in the CDMS, eligibility is confirmed and signed informed consent is obtained, patients will be assigned to the next sequentially coded intervention pack that is available at the recruiting site. Patients are randomised 1:1:1:1 into one of four groups: Metamizole +educational intervention versus metamizole +standard care versus ibuprofen +educational intervention versus ibuprofen +standard care. The intervention packs will contain a pill bottle and an envelope with the assignment of the short versus no intervention. The intervention packs will be prepared according to the randomisation list by an independent hospital pharmacist not involved in the study.

### Blinding

Staff at the hospital pharmacy not involved in patient recruitment or other trial activities will fill the gelatin capsules with the study medication and label the pill bottles according to a standard operating procedure and based on the randomisation list provided by the CTU Bern. The study medication will be delivered in identically labelled sealed pill bottles containing capsules of identical appearance. The randomisation list will be kept electronically at the CTU Bern in a password-protected file. Study personnel and physicians involved in patient recruitment, treatment and outcome assessment will have no access to the list.

Due to the nature of the short educational intervention, no blinding of patients receiving an information leaflet and a telephone call is possible. Therefore, contamination and performance bias are important to consider. The envelope including the leaflet will be sealed and the patients will be instructed to open de envelope outside the GPs office. GPs will not be informed about the group assignment and the content and aim of the educational intervention. Patients assigned to the educational intervention will receive a telephone call according to a predefined script by a member of the study team otherwise not involved in patient recruitment, outcome assessment and data analysis. To minimise the risk of detection bias, the study outcomes are collected in pain diaries and self-reported questionnaires (NRS at day 14, COMI at day

42). Study personnel that will contact patients by phone for reminders and to collect information on AEs will be blinded to the group assignment. To assess differences in additional treatments (performance bias), patients and GPs will record additional treatments other than the study medication and the educational information (eg, additional medication use, physical therapy, injections).

The statistician responsible for the final analysis will also have no access to the list until the primary analysis of the trial is finished. Data analysis will be done according to a prespecified analysis plan before the randomisation code will be broken. Patients, physicians and data analysts will be blinded with regard to the allocation to the study medication until after the primary analysis. To assess the success of blinding, we will ask patients at the end of the study into which group they belief they had been assigned to. Data on all recruited patients will be collected and regularly monitored.

## Statistical analysis plan

The statistical analysis will be performed in compliance with the International Conference on Harmonisation's Good Clinical Practice guidelines and the report will be developed in line with the Consolidated Standards of Reporting Trials statement.

To assess the efficacy for the two primary outcomes, we will use two analysis sets (see figure 1). The full analysis set (FAS) includes all randomised patients for the intention-to-treat (ITT) analysis, that is, all patients will be analysed in the group they were assigned to. The per-protocol (PP) set is based on the FAS, excluding patients with major protocol violations, patients who were excluded due to the predefined stop criteria, patients withdrawing from the trial within 3 days of treatment start, patients undergoing surgery and patients that did not receive the randomised treatment or intervention.

We will present patient and procedural characteristics using descriptive statistics (mean and SD or median and IQR for continuous variables and frequency and percentage for categorical variables).

To compare the efficacy of metamizole and ibuprofen (non-inferiority), we use an ITT (FAS set) and a PP analysis (PP set) for the primary outcome (NRS change from baseline to follow-up at 14 days). Both analyses need to meet non-inferiority in order to claim success. The difference in the change in NRS between the two groups will be calculated from a linear model, adjusted for the baseline NRS value and the type of intervention (short intervention vs standard care). If the lower one-sided 97.5% confidence limit of the difference lies above the non-inferiority margin of −1, non-inferiority will be claimed. If we can demonstrate non-inferiority, we will additionally test for superiority at a two-sided significance level of 2.5%. Since testing is nested (ie, hierarchical), this procedure has no implication on the overall type-I error rate.

To compare the primary outcome between the short educational intervention to usual care (superiority), we use ITT analysis (COMI change from baseline to day 42).

The difference in the change in COMI between the two groups will be calculated and tested at a two-sided significance level of 2.5% using a linear model, adjusted for the baseline COMI and the type of drug treatment. Differences will be presented with a two-sided 97.5% CI and the regular 95% CI.

We will assess both primary outcomes for interaction. We will introduce an interaction term between the treatment indicator and the intervention indicator to test whether in (A) the effect of the metamizole treatment on the change in NRS at day 14 is different in patients that receive the short intervention compared with the standard care group and in (B) whether the effect of the short intervention on the change in COMI at day 42 is different in patients that receive metamizole compared with patients that receive ibuprofen.

Secondary continuous outcomes will be analysed as the primary outcomes. Besides the indicator for the drug treatment or intervention, models will be adjusted for the baseline value if available. Secondary binary outcomes will be analysed using logistic regression models adjusted for the indicator for the drug treatment or intervention, as applicable. Time-to-event outcomes will be analysed using Cox regression adjusted for the indicator for the drug treatment or intervention, as applicable. For all secondary effect measures, we will display regular two-sided 95% CI and accompanying p values.

Further, we will perform additional PP analyses for the primary outcome of the comparison between the two intervention arms as well as for all secondary outcomes. If outcomes are missing, we will employ multiple imputation in the primary analysis and additionally perform an available case analysis disregarding missing data. Continuous outcomes that are measured several times during follow-up will additionally be evaluated in a repeated measures mixed-effects linear model, additionally introducing a random intercept for patients into the model.

Exploratory analysis will be conducted with regards to predictors for treatment responses. We will explore the association between clinical predictors (eg, depression, fear avoidance beliefs, social factors), drug levels, or genetic mutations and the treatment response in linear models. We will perform subgroup analyses according to gender, genetic phenotypes and drug levels.

Safety outcomes will be reported as proportion and 95% CI separately for the two drug treatment groups using the safety population (ie, all patients that have received at least one dose of the study drug and at least one safety-related visit or observation) set.

Patients with missing outcome data will be imputed using multiple imputation.

## Safety

Protocol violations should not lead to treatment discontinuation unless they indicate a significant risk to patient safety.

Patients may be withdrawn from the study treatment for the following reasons:

► Haematological disorders including leucopaenia (defined as leucocyte count <3.0 G/L; thrombocytopaenia (defined as <100 G/L) and anaemia (decrease in haemoglobin ≥20 g/L or <100 g/L).

► Potential evidence for a haematological disease (pathological blood smear needing further evaluation).

► Decreased kidney function (defined as an estimated creatinine clearance using the MDRD formula of <50 mL/min/1.73 m$^2$) or AKI stage 1 or higher (defined according to the KDIGO definition: ≥1.5 times creatinine increase compared with baseline value or ≥26.5 μmol/L creatinine increase).

► Development of toxicity which, in the investigators or GPs judgement, precludes further use of one of the study drugs.

► Patients lost to follow-up or non-compliance.

We will comply with all regulations concerning safety measures in clinical trials as set forth by the Swiss Agency for Therapeutic Products (Swissmedic). The investigators will report any serious AEs occurring during clinical trials, independent of direct causal relationship with the treatment, within 24 hours. In case of a suspected unexpected serious adverse reaction, Swissmedic will be notified by the sponsor–investigator within the legal timelines. Unblinding will be permissible in case the information is required to ensure the patients safety in case of an AE.

All AEs are reviewed by a safety committee and graded according to the National Health Service (NIH) Common Terminology Criteria for Adverse Events (CTCAE V.5.0).[54]

### Patient and public involvement

In this study, we use information material developed by patient advocacy groups. The results of this study will be communicated to the patients and physicians participating in the trial. The results will also be communicated to patients' advocacy groups and may help to improve the information material. Further, we will assess the impact of the study on patients and factors that may be facilitators or barriers to improve pain and pain management in LBP during in-person interviews.

### Clinical implications

During an LBP episode treatment guideline recommend to alleviate pain using pain medication and to stay physically active. To alleviate pain, NSAIDs are frequently used. However, the use of NSAIDs may be limited because of common adverse effects. Metamizole has a favourable adverse effects profile and may be used as an alternative. However, there is a lack of data on its effectiveness and safety for the treatment of LBP. This is the first study that systematically investigates the efficacy of metamizole in LBP patients undergoing treatment over the course of several weeks. Therefore, this study will provide evidence to support or discourage the use of metamizole in LBP. The findings of this study will be especially relevant for those patients where NSAIDs should not be used. Moreover, this trial is expected to provide evidence for a highly needed non-opioid treatment alternative in

musculoskeletal pain. In particular, because the use of opioids has been shown to be no more effective than non-opioid medication[55] and is associated with potentially severe AEs.

Non-pharmacological educational measures may potentiate the effect of the pain medication in acute LBP by addressing misbeliefs and avoiding disuse. Despite good evidence that staying active during an acute LBP episode is more effective than bed rest,[56] more than 50% of the Swiss population report avoiding movements and increasing rest during an LBP episode.[57] The interaction between patient and doctor early on influences the recovery of patients and many patients may benefit from a short educational intervention that is easy to be implemented even in a busy GP practice in addition to pain medication. This is the first study that investigates the impact of a short educational intervention in addition to pain medication on the recovery and medication adherence in patients with acute and subacute LBP. Even a small individual effect may have a large impact on the overall societal burden of LBP. Our work will inspire novel approaches on how to improve adherence to medication and behavioural changes.

### Ethics and dissemination

The trial will be performed in compliance with the Declaration of Helsinki and its amendments, and has been approved by the responsible Ethics Board (Ethikkommission Bern/2018-01986) and the Swiss Agency for Therapeutic Products (Swissmedic/2019DR4002). Important protocol modifications will be communicated to the Ethics Board and participating study sites. Potential participants must provide written informed consent before entering the study. Subjects can leave the study at any time for any reason without any negative consequences for their further treatment. When a patient ends the treatment phase of the study prematurely, we will record the date and reason for early treatment discontinuation. If possible, the end of treatment evaluations will be collected before the patient is started on any other therapeutic intervention. Insurance coverage will be provided for all study participants by the study sponsor.

The results of the main trial and each of the secondary outcomes will be submitted for publication in open access peer-reviewed journals. Results will be disseminated to all participating centres and participants expressing interest. The EMISI study group is an independent academic research group, which will not employ professional writers. The EMISI study group will comply with the open access regulation of the Swiss National Science Foundation.

**Author affiliations**
$^1$Department of General Internal Medicine, University of Bern, Bern, Switzerland
$^2$Department of General Internal Medicine, Bern University Hospital, Bern, Switzerland
$^3$Institute of Primary Health Care (BIHAM), University of Bern, Bern, Switzerland
$^4$CTU Bern, University of Bern, Bern, Switzerland
$^5$Department of Psychology, University of Bern, Bern, Switzerland

[6]Department of Clinical Pharmacology and Toxicology, University of Bern, Bern, Switzerland

**Contributors** MMW, SS, AL, JSF and MH designed the study. A-NF, VS and CR critically commented on the methods and contributed to the development of the study. MMW and JSF wrote the first draft of the manuscript. All authors revised the manuscript and approved the final version.

**Funding** Efficacy of Metamizole versus Ibuprofen and a Short Educational Intervention versus Standard Care in Acute and Subacute Low Back Pain: A Randomized, Factorial Trial (The EMISI Trial) is supported by the Swiss National Science Foundation (grant number: 32003B-179346/1).

**Disclaimer** This research project received no further support from any other funding agency.

**Competing interests** MH and MMW received a grant from the Swiss Science National Foundation for the study: grant number 32003B_179346. The funder plays no role in the study design, collection of data, analysis, and writing of the manuscript. There is no conflict of interest regarding intellectual content and proprietary affairs.

**Patient and public involvement** Patients and/or the public were involved in the design, or conduct, or reporting, or dissemination plans of this research. Refer to the Methods section for further details.

**Patient consent for publication** Consent obtained directly from patient(s)

**Provenance and peer review** Not commissioned; externally peer reviewed.

**ORCID iDs**
Maria M Wertli http://orcid.org/0000-0001-6347-0198
Sven Streit http://orcid.org/0000-0002-3813-4616

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
