## [Reviewer comments · BMJ Open]

ARTICLE DETAILS

TITLE (PROVISIONAL)	Efficacy of Metamizole versus Ibuprofen and a Short Educational Intervention versus Standard Care in Acute and Subacute Low Back Pain: A Study Protocol of a Randomized, Multicenter, Factorial Trial (EMISI Trial)
AUTHORS	Wertli, Maria M; Flury, Julian S.; Streit, Sven; Limacher, Andreas; Schuler, Vanessa; Ferrante, Asha-Naima; Rimensberger, Caroline; Haschke, Manuel

VERSION 1 – REVIEW

REVIEWER	van Uum, Rick University Medical Center Utrecht, Primary Care & Health Sciences
REVIEW RETURNED	30-Mar-2021

GENERAL COMMENTS	Reviewer: Rick T. van Uum, MD PhD, GP trainee The authors present the study protocol of a multicenter, randomized, double-blind trial with a factorial design to compare the effectiveness of ibuprofen, metamizole and short-intervention for treatment of acute lower back pain. In general, I think the study is well designed and the study protocol is well written. It has the potential to provide answers to a very relevant complaint frequently encountered in clinical practice. I do, however, have some methodological concerns/questions. INTRODUCTION In many countries, paracetamol (despite its ineffectiveness for treating LBP) is first choice for acute lower back pain, rather than ibuprofen. The authors' focus on ibuprofen limits the generalisability of the study. Please reflect on why paracetamol hasn't been included in the study design, and discuss the limitation of this choice. Dutch guidelines recommend a focus on encouraging physical activity of the patient, with only a supporting role for pain killers. This is based on limited effectiveness (with questionable clinical relevance) of paracetamol and/or NSAIDs for acute LBP. See: • Williams CM, et al. Efficacy of paracetamol for acute low-back pain: a double-blind, randomised controlled trial. Lancet, 2014.• Roelofs PDDM. Non-steroidal anti-inflammatory drugs for low back pain. Cochrane, 2008, CD000396 Please tone down the effectiveness of NSAIDs in the introduction section. Please be specific on the risk of agranulocytosis. Provide numbers.
--

	ELIGIBILITY CRITERIA The exclusion criteria are very extensive, potentially limiting trial recruitment and future generalisability. For example, there a many patients with a previous history of anemia, in many cases not predisposing for adverse effects of study medication. Is it possible further specify or limit the exclusion criteria and if not, how will the authors handle possible difficulties in recruitment? SETTING Authors will recruit patients from outpatient clinics, GP practices and emergency departments. These are in certain aspects very different settings with different types of patients and acute LBP episodes. Although it may make the results of the study widely applicable, it may also prove impractical in interpreting study results. How are authors going to address this issue in their analysis and wouldn't it be more preferable to limit the study to one specific setting? SAMPLE SIZE AND POWER To me, it is unclear whether the current sample size calculation takes into account the factorial design of the trial. It seems as though the study is powered on both interventions (medication – education) separately, but shouldn't it be combined, resulting in a higher number of patients needed to provide meaningful conclusions? This section is confusing to read. Consider rearranging the paragraphs and moving the first paragraph to become the last. Drop line 34-36. EXPLANATORY VARIABLES Study procedures include drawing blood to assess drug plasma concentrations and genetic polymorphisms. However, the authors do not explain specifically how these results will be necessary or useful for interpretation, nor are these outcomes mentioned as secondary outcomes. Please be specific on how these blood tests will guide the interpretation of study results. STUDY PROCEDURES P 16, line 11-18: which characteristics, questionnaires and clinically assessed details? P 16, line 21-31: this procedure seems highly precautionary to me and conflicts with the otherwise pragmatic design of the trial (with multiple settings, adaptable dosing, etcetera). Why is this precaution necessary, concerning the low risk of severe AE and already available patient history (i.e. I assume that in outpatient clinics and GP practices kidney function etc is known in many patients). PHARMACOLOGICAL INTERVENTION The adaptable dosing schedule is well-thought of, and allows for good comparison with clinical practice. My compliments. CONCOMITANT INTERVENTIONS Physicians are discouraged, but allowed, to provide rescue medication and PT advice. How are the authors going to collect data on this? ETHICS
--	---

	I am left with the question whether the number of invasive procedures (multiple blood drawing, genetic testing, kidney function, anemia) proportionate to the research question and clinical practice? If metamizole is so routinely used in many countries (in the Netherlands experience is limited), why are so many blood counts necessary to monitor potential AEs? This gives me, as reviewer, the hunch that metamizole might not be as safe as proposed. Please elaborate on the safety profile of metamizole.
--	--

REVIEWER	Lapi, Francesco Health Search, Italian College of General Practitioners and Primary Care, Epidemiology
REVIEW RETURNED	27-Apr-2021

GENERAL COMMENTS	Overall, the protocol is clear and well written. However, some details should be added to better understand the study aim. Metimazole and ibuprofen are old medications. Why do the authors would compare their efficacy? Some aspects on metimazole-related advantage should be added. For instance, there is a cost-effectiveness advantage as well? Why did the authors chose ibuprofen as comparator? LBP being selected in primary care might differ from those identify in the emergency department. Are there analyses on this aspect? In this respect, multilevel modeling has been considered? Effect modification due to edutional intervention should be also tested? Clinical implication of the expected results should be discussed more in depth.
--

VERSION 1 – AUTHOR RESPONSE

Reviewer #1 Mr. Rick van Uum, University Medical Center Utrecht

Comments to the Author: The authors present the study protocol of a multicenter, randomized, double-blind trial with a factorial design to compare the effectiveness of ibuprofen, metamizole and short-intervention for treatment of acute lower back pain. In general, I think the study is well designed and the study protocol is well written. It has the potential to provide answers to a very relevant complaint frequently encountered in clinical practice. I do, however, have some methodological concerns/questions.

Comment 1: INTRODUCTION

In many countries, paracetamol (despite its ineffectiveness for treating LBP) is first choice for acute lower back pain, rather than ibuprofen. The authors' focus on ibuprofen limits the generalisability of the study. Please reflect on why paracetamol hasn't been included in the study design, and discuss the limitation of this choice.

Response: Thank you for this comment. Although paracetamol has been recommended due to its favorable side effect profile, hepatotoxicity is a matter of concern. After the publication of one RCT that showed paracetamol to be no more effective than placebo (William, Lancet 2014), we believe that a comparison of metamizole with paracetamol does not add significant new knowledge as we expect metamizole to be as effective as NSAIDs. Current treatment guidelines consistently recommend non-steroidal anti-inflammatory drugs (NSAIDs) as the first pharmacological choice (Foster, Lancet 2018; Quaseem, Annals Int. Medicine 2017; van Tulder, Europ Spine J 2006; Chou, Annals int. Medicine 2017, Wong, Europ J Pain 2017). The aim of this study was to compare the efficacy of metamizole to the most commonly used NSAID in Switzerland (which is ibuprofen, see also Wertli, BMC Health

Services Research 2017). As such, we believe that the findings will be generalizable to other countries where NSAIDs are frequently used and often available over the counter.

Comment 2: Dutch guidelines recommend a focus on encouraging physical activity of the patient, with only a supporting role for pain killers. This is based on limited effectiveness (with questionable clinical relevance) of paracetamol and/or NSAIDs for acute LBP. See:

- Williams CM, et al. Efficacy of paracetamol for acute low-back pain: a double-blind, randomised controlled trial. *Lancet*, 2014.
- Roelofs PDDM. Non-steroidal anti-inflammatory drugs for low back pain. *Cochrane*, 2008, CD000396

Please tone down the effectiveness of NSAIDs in the introduction section.

Response: We agree, that non-pharmacological measures should be the first choice. However, when patients seek care from their primary care physician or in the ER, pain medications usually are needed to alleviate acute pain. We rephrased the introduction accordingly (page 5, paragraph 2). Compared to other pain medications, there are randomized trials available that showed the efficacy of NSAIDs (Friedman, *Jama*. 2015; Rasmussen-Barr E, *Cochrane Review* 2016). But we agree with the reviewer that many studies have methodological limitations. This is one reason why we designed the current study.

Comment 3: Please be specific on the risk of agranulocytosis. Provide numbers.

Response: the incidence of agranulocytosis is now discussed in the revised Introduction (page 5 and 6, paragraph last and first paragraph).

Comment 4: ELIGIBILITY CRITERIA

The exclusion criteria are very extensive, potentially limiting trial recruitment and future generalisability. For example, there are many patients with a previous history of anemia, in many cases not predisposing for adverse effects of study medication. Is it possible further specify or limit the exclusion criteria and if not, how will the authors handle possible difficulties in recruitment?

Response: Thank you for this comment. Because no previous study has assessed the efficacy of metamizole in musculoskeletal pain and only limited studies were performed over a longer follow-up duration, the main goal was to minimize the risk for patients. As most exclusion criteria are similar for NSAIDs and metamizole, we do not feel that the exclusion criteria will affect recruiting. Indeed, first months of recruiting show that recruiting is not limited by the exclusion criteria but because of COVID 19 pandemic and other factors. The only exclusion criteria that was misleading was "current opioid use". Originally, we intended to exclude patients with chronic preexisting opioid use. However, patients that were temporarily prescribed opioids may also be included in the study. In the meantime the exclusion criteria was updated to "chronic opioid use" and submitted to the ethical committee. Certain exclusion criteria (e.g. pregnancy test) were mandatory by the Swiss regulatory authorities (Swissmedic and the Ethical committee).

Comment 5: SETTING

Authors will recruit patients from outpatient clinics, GP practices and emergency departments. These are in certain aspects very different settings with different types of patients and acute LBP episodes. Although it may make the results of the study widely applicable, it may also prove impractical in interpreting study results. How are authors going to address this issue in their analysis and wouldn't it be more preferable to limit the study to one specific setting?

Response: We agree with the reviewer that the study setting may influence the study findings. Randomization was stratified by center and we will account for the study setting in the multivariate model. We did not include this information in the study protocol to conceal treatment allocation.

Comment 6: SAMPLE SIZE AND POWER

To me, it is unclear whether the current sample size calculation takes into account the factorial design of the trial. It seems as though the study is powered on both interventions (medication – education) separately, but shouldn't it be combined, resulting in a higher number of patients needed to provide meaningful conclusions?

Response: We expect the two primary hypotheses to be independent of each other and do not expect an interaction between both interventions. Therefore, we powered each hypothesis individually, but accounted for multiple testing by adjusting the type-I-error rate by setting the alpha level to 0.025 in each test. We now discuss this in the revised sample size calculation to improve clarity (page 9 and 10, paragraph 3 and 1).

Comment 7:

This section is confusing to read. Consider rearranging the paragraphs and moving the first paragraph to become the last. Drop line 34-36.

Response: We were unsure which section the reviewer referred to. We revised several sections of the methods section to improve readability. We hope we have addressed the reviewer's comment.

Comment 8: EXPLANATORY VARIABLES

Study procedures include drawing blood to assess drug plasma concentrations and genetic polymorphisms. However, the authors do not explain specifically how these results will be necessary or useful for interpretation, nor are these outcomes mentioned as secondary outcomes. Please be specific on how these blood tests will guide the interpretation of study results.

Response: We thank the reviewer for this comment. Various pharmacokinetic and pharmacodynamics factors may influence treatment response to pain medication. This is one of the first studies that aims to identify potential factors that influence treatment response to ibuprofen and metamizole. We added this aspect in the revised methods section (page 13, line 2).

Comment 9: STUDY PROCEDURES

P 16, line 11-18: which characteristics, questionnaires and clinically assessed details?

Response: We added the questionnaires and characteristics including the appropriate references for validation studies (page 14, paragraph 2). All questionnaires that are collected are additionally included in Table 1.

Comment 10: P 16, line 21-31: this procedure seems highly precautionary to me and conflicts with the otherwise pragmatic design of the trial (with multiple settings, adaptable dosing, etcetera). Why is this precaution necessary, concerning the low risk of severe AE and already available patient history (i.e. I assume that in outpatient clinics and GP practices kidney function etc is known in many patients).

Response: We understand the reviewer's point. We pragmatically refrained from including lab values as exclusion criteria because we assume that most patients eligible for the study will not have exclusion criteria. However, NSAIDs and metamizole may have potential serious adverse effects and we (and the ethical committee) felt it to be appropriate to add these tests as a safety measure. The blood samples will be covered by the study budget and patients are informed about the need to analyze blood. Patients can decline in case they do not wish to have their blood analyzed. However, most patients appreciate this approach. We now added as a supplementary file the patient consent form.

Comment 11: PHARMACOLOGICAL INTERVENTION

The adaptable dosing schedule is well-thought of, and allows for good comparison with clinical practice. My compliments.

Response: Thank you for this positive comment.

Comment 12: CONCOMITANT INTERVENTIONS

Physicians are discouraged, but allowed, to provide rescue medication and PT advice. How are the authors going to collect data on this?

Response: Physicians will record prescription(s) of additional pain medications and PT in their intake forms. Further, patients should record all treatments in their pain diary. During the follow-up calls and visits, patients will be asked whether they were taking additional medications and / or had other treatments. We included this information in the revised Methods (page 15, 16, and 18; paragraph 2, 1, and 1, respectively).

Comment 13: ETHICS

I am left with the question whether the number of invasive procedures (multiple blood drawing, genetic testing, kidney function, anemia) proportionate to the research question and clinical practice? If metamizole is so routinely used in many countries (in the Netherlands experience is limited), why are so many blood counts necessary to monitor potential AEs? This gives me, as reviewer, the hunch that metamizole might not be as safe as proposed. Please elaborate on the safety profile of metamizole.

Response: Metamizole is an old medication that has never been studied in clinical trials. Although the safety profile is overall good (see also in the revised Introduction), we don't know why some

patients may develop agranulocytosis. While this rare adverse event is not expected to occur during the current study, we aim to elucidate potential underlying factors. Further, the aim of the study is also to shed some light into individual pharmacokinetic and pharmacodynamics of pain medications. While these analyses are exploratory, we aim to advance our knowledge on the treatment of acute pain beyond the primary goal of our study. We believe by using vigorous methods we will be able to achieve these goals that then need to be confirmed in future studies. Patients will be informed about these additional exploratory analyses and are free to consent. We clarified the explanatory nature of these analyses in the appropriate sections.

Reviewer #2:

Dr. Francesco Lapi, Health Search, Italian College of General Practitioners and Primary Care
 Comments to the Author:

Comment 1: Overall, the protocol is clear and well written. However, some details should be added to better understand the study aim. Metimazole and ibuprofen are old medications. Why do the authors would compare their efficacy? Some aspects on metamizole-related advantage should be added. For instance, there is a cost-effectiveness advantage as well?

Response: We thank the author for this comment. Although both medications are old, no study assessed the efficacy of metamizole in musculoskeletal pain. Based on clinical experience, we expect to find a similar efficacy as ibuprofen. However, we don't know whether this is truly the case. There is no cost advantage of metamizole compared to ibuprofen. Both are readily available and cheap. We included examples for more favorable safety profile compared to NSAIDS and paracetamol in the revised Introduction (page 5, paragraph 3).

Comment 2: Why did the authors chose ibuprofen as comparator?

Response: The aim of this study was to compare the efficacy of metamizole to the most commonly used NSAID in Switzerland (which is ibuprofen, see also Wertli, BMC Health Services Research 2017). After the publication of one RCT that showed paracetamol to be no more effective than placebo (William, Lancet 2014), we believe it not appropriate to choose paracetamol as a comparator. Clinical experience indicates that metamizole is equally effective than ibuprofen. In case we will be show non-inferiority, clinicians will have an additional non-opioid pain medication to choose from.

Comment 3: LBP being selected in primary care might differ from those identify in the emergency department. Are there analyses on this aspect? In this respect, multilevel modeling has been considered? Effect modification due to educational intervention should be also tested?

Response: We agree with the reviewer that the study setting may influence the study findings. Randomization was stratified by center and we will account for the study setting in the multivariate model. We did not included this information in the study protocol to conceal treatment allocation.

Comment 4: Clinical implication of the expected results should be discussed more in depth.

Response: We now include a Clinical Implication section (page 23, paragraph 2).

VERSION 2 – REVIEW

REVIEWER	van Uum, Rick University Medical Center Utrecht, Primary Care & Health Sciences
REVIEW RETURNED	04-Jul-2021

GENERAL COMMENTS	The authors have provided a satisfactory response to my previous review comments and have adjusted the manuscript accordingly. I have one remaining remark. Although paracetamol is not as effective as NSAIDs, and I appreciate the impossibility of adapting the study protocol to include a paracetamol arm, this remains a limitation to the generisability of the findings. I would recommend authors to include some lines to discuss this limitation.
--

VERSION 2 – AUTHOR RESPONSE

Reviewers' comments:

Reviewer #1 Mr. Rick van Uum, University Medical Center Utrecht

The authors have provided a satisfactory response to my previous review comments and have adjusted the manuscript accordingly.

I have one remaining remark. Although paracetamol is not as effective as NSAIDs, and I appreciate the impossibility of adapting the study protocol to include a paracetamol arm, this remains a limitation to the generalisability of the findings. I would recommend authors to include some lines to discuss this limitation.

Response:

We agree with the reviewer that selected patient sample may be a threat to the generalizability of study results. However, studies of specifically defined groups may also generalize to extend our knowledge (Kukull WA, Neurology. 2012). Generalizability refers to the “big picture” interpretation of a study's results once they are determined to be internally valid.

Although paracetamol may be frequently used in many countries, evidence suggests that it is no more effective than placebo in patients with acute low back pain (William, Lancet 2014). From the clinical experience of the principal investigator and the research group, we disagree that paracetamol is the first choice in acute low back pain in the clinical setting that we aim to recruit patients. However, paracetamol is often used in situations where NSAIDs cannot be used due to contraindications. This is exactly the research question the study aims to address. The findings will be of high relevance because it will answer whether metamizole is equally or less effective than NSAIDs.

Considering that the current study aims to address two specific research questions, adding another research question to the study would affect the internal validity and the feasibility of the study. From a clinical and scientific perspective, it would be unethical to add a paracetamol arm to this study. Studies have shown, that NSAIDs are effective in acute low back pain (e.g., Friedman BW, Jama). Based on this evidence, treatment guidelines consistently recommend NAIDs as the first pharmacological choice (Foster, Lancet 2018; Quaseem, Annals Int. Medicine 2017; van Tulder, Europ Spine J 2006; Chou, Annals int. Medicine 2017, Wong, Europ J Pain 2017). We recognize, that clinical practice in the Netherlands may be different but feel this is not sufficient to see it as a limitation to the overall generalizability. Thus, we do not feel that adding a limitation to the study protocol will increase the quality of the (already very long) manuscript. However, depending on the study results it may be appropriate to address this aspect when we publish the results of the study.

Thank you again for your comments. We thank the reviewer for their very helpful comments and believe that the revisions improved the manuscript. We hope that our revised paper is now suitable for publication.